# How to Recognize Animals’ Vulnerability: Questioning the Orthodoxies of Moral Individualism and Relationalism in Animal Ethics

**DOI:** 10.3390/ani10020235

**Published:** 2020-02-02

**Authors:** Martin Huth

**Affiliations:** Messerli Research Institute, Unit Ethics and Human-Animal Studies, Veterinarplatz 1, 1210 Vienna, Austria; martin.huth@vetmeduni.ac.at

**Keywords:** moral individualism, relationalism, vulnerability, recognizability, immanent critique

## Abstract

**Simple Summary:**

Many animal ethicists consider cognitive capacities as being the basis for the moral status of an animal. On this view, animals that have, for instance, complex experiences, future preferences, or at least the ability to suffer, impose an obligation on us. Those beings that do not share these capacities do not have a moral status. This would also apply to embryos, infants, or severely cognitively impaired humans, but this seems to be at odds with many of our shared ethical intuitions. As a response, so-called relationalists argue that our different relations to different kinds of beings form the basis for moral obligations. However, on this view, it remains unclear (a) why it is particularly our relations to kinds of animals that are morally relevant; and (b) how we can criticize and change these relations. This paper seeks to combine both accounts of animal ethics to overcome these pitfalls. It argues that it is individual vulnerability that forms the basis of moral obligations, but that social structures and relations pre-determine how we perceive and recognize vulnerability. However, particular relationships with animals as well as open possibilities to treat animals in different ways (e.g., to treat a dairy cow not as a mere resource) render critique and change of current practices possible.

**Abstract:**

This paper presents vulnerability and the social structures surrounding *recognition of vulnerability* as fundamental elements of animal ethics. Theories in the paradigm of *moral individualism* often treat individual rational capacities as the basis of moral considerability. However, this implies that individuals without such capacities (such as human or nonhuman infants) are excluded even though we grant them special protection in our lived morality. It also means that moral agents are pictured as disembodied, impartial observers, independent of social relations and particular relationships. *Relationalists* take moral obligations to be rooted in different *kinds* of beings. However, relationalism runs the risk of losing the individual animal and her capacities. It cannot also it adequately explain what forms different kinds of being, or the constitution of normativity through relations. Moreover, it lacks resources to explain how critique and change are possible. This paper argues that vulnerability and its recognition are the source of our moral obligations to animals. It seeks to combine individualist and relationalist arguments by using a social ontology of the bodily individual which can be applied to human agents and to any vulnerable being. Social structures predetermine the ways in which we perceive and recognize the vulnerability of living beings. However, we are not fully determined by these structures; particular relationships and direct encounters with individual animals as well as the open possibilities of treating animals differently that are immanent in common practices render critique and change of current conditions possible.

## 1. Introduction

The aim of this paper is to identify vulnerability combined with the recognition of vulnerability and its social prerequisites as a central concept of animal ethics. The considerations presented below draw on recent debates in feminist ethics, poststructuralism, and phenomenology, where vulnerability has attracted a good deal of attention, particularly in the past 15 years—as Judith Butler’s *Precarious Life* [1], the anthology *Vulnerability. New essays in Ethics and Feminist Philosophy*, edited by Catriona Mackenzie, Wendy Rogers and Susan Dodds [2], and Erinn Gilson’s *The Ethics of Vulnerability* [3] illustrate. Particularly Butler’s poststructuralist theory of *recognizability*, developed in her *Frames of War* [4], is an important point of reference and will be clarified later in the paper.

A common goal of these authors is to contest a problematic presupposition, which can be found in canonical ethical theories, including those of deontological, utilitarian, and contractarian character, and detected in animal ethics, too. Proceeding from the (Aristotelian) axiom that equals should be treated equally, “orthodox” authors in practical philosophy tacitly or intentionally rely on a modern, liberalist ontology of the self-sufficient (non-)human subject and contend that individuals who share particular rational capacities (such as self-awareness or the ability to have conscious experiences of suffering) have to be considered equally. Dedicated *moral individualists* such as James Rachels [5] and Jeff McMahan [6] stress that the equal consideration in question should be independent of species or group membership. Starting from this cluster of assumptions, animal ethicists often emphasize that animals are competent rational beings which merit moral consideration and should not be reduced to “biomachines” [7]. However, this ratiocentric perspective often generates counter-intuitive consequences by excluding individuals who appear to deserve special protection in our lived morality (e.g., infants or cognitively disabled individuals). In contrast, philosophers such as Butler [1] and Gilson [3] identify corporeality (beyond the dualism of body and mind), and its immanent vulnerability, as an especially important source of moral obligations. Vulnerability is conceived of, not as a merely intrinsic property (although it must be understood as an individual’s vulnerability), but as something emerging through corporeal beings’ inevitable dependency on, and relatedness to, others. Although authors such as Butler and Gilson frequently emphasize that their approach is applicable to animals [3,4], they often decline to explain exactly how. 

Recently, several animal ethicists, such as Angela Martin [8], Angie Pepper [9], and Corine Pelluchon [10], have drawn on the concept of vulnerability. A few writers, including Stephen Thierman [11] and Ani Satz [12], had made use of the concept earlier still. However, the concept is understood in remarkably different ways:(a)Martin, an analytical philosopher, takes vulnerability to be based on the individual’s agency- and welfare-related interests; she therefore closely links vulnerability to sentience [8] (p. 2). *Particular* vulnerability emerges when prejudices increase the probability that the vulnerable individual’s interests will be considered unjustly [8] (p. 2; 11). While vulnerability is *prima facie* an individual property, particular (i.e., heightened) vulnerability emerges in unjust social relations or exploitative relationships.(b)Adopting a perspective lying between ethics and politics, Pepper identifies increased vulnerability with the reification of animals as trading goods, the ignorance of a global network of human–animal relations, and the exclusion of nonhuman beings from our notions of justice. The acknowledgment of animals’ subjectivity and recognition of their complex and ubiquitous interrelations become crucial ethical and political issues [9].(c)Drawing on the work of the French phenomenologist Emmanuel Levinas, Pelluchon understands vulnerability as the source of responsibility. By virtue of our vulnerability we are addressed by the nonhuman other—*prima facie* independently of their particular cognitive capacities [10]. Vulnerability is thus understood as dependence and, at the same time, as an ethical power that makes a moral claim going beyond the consideration of individual capacities and interests.

Moreover, some authors conceive of vulnerability as a feature that is complementary to sentience or welfare. Clare Palmer [13] considers sentience the basis of an individual’s moral status; vulnerability is rooted in dependency arising from asymmetrical relationships between humans and animals. Similarly, Elisa Aaltola [14] argues that sentience is precisely what renders living beings vulnerable. However, she argues against interpreting animals as purely passive, and—in contrast to authors such as Butler or Gilson—opposes vulnerability to animal subjectivity and activity or agency [14] (p. 134). 

We can see, then, that this concept has been used quite heterogeneously and from different perspectives. In what follows, I will mainly draw on Butler’s reflections in *Frames of War* [4], together with the established phenomenology of the body, in proposing an account of animal ethics that is rooted in bodily vulnerability and its social recognition. In my view, vulnerability is rooted in a social ontology of corporeality. Corporeality has to be associated with dependence on resources (nutrition, water, habitats, intact ecosystems, etc.) and relatedness to other living beings (even if it is an unwilled proximity) [4] (p. 34)—therefore, corporeality implies vulnerability. Vulnerability is described as that what “addresses” us as moral agents. As even ignorance of vulnerability must be understood as a response to it, we cannot but be responsive to it. Similarly, it is introduced as what forms the basis of moral obligations [1,3,4,10], since it compels us to determine justifiable ways of responding. 

At the same time, we must acknowledge that there is no vulnerability per se. In other words, vulnerability is inevitably framed—but not entirely determined or constructed—by social structures. Generally speaking, we experience animal vulnerability differently when we encounter a “charming” puppy, a “disgusting” sewer rat (and zoonotic vector), or a “noble” wild animal such as an eagle; however, any particular relationship or encounter might not only reproduce, but also exceed and disrupt these frames (see below). Vulnerability is therefore conceptually and inextricably connected with: (a) overarching *social relations* that sustain its recognition; and (b) *particular relationships* and encounters in which responsibility (which can be expressed in care and respect, or in abuse and exploitation) can be traced back to one individual’s responsiveness to another’s vulnerability, which is itself pre-determined, but not determined by social relations and frames.

This conceptualization of vulnerability as embedded and emergent in relations and relationships undermines the alleged dualism between moral individualism and relationalism. In Section 2, I critically analyze moral individualism. This has been the most influential starting point (the actual orthodoxy) in animal ethics for almost five decades now and is represented by Rachels [5], McMahan [6], and of course the “classical” animal ethicists Peter Singer [15] and Tom Regan [16]. In Section 3, I focus on relationalism—a second thread in the debate that has received less attention and that builds the counterpart of moral individualism ever since the consolidation of the discipline, as Diamond’s text *Eating Meat and Eating People* [17] published in 1978 documents. Predominantly, these theories employ the *argument from kind*, i.e., the claim that our relations to animals are primarily relations with particular kinds of being, and the further claim that this has normative implications regarding the treatment of individuals (whereas individual capacities, but also vulnerabilities, seem to play a subordinate role). In Section 4, drawing on Emmanuel Levinas’ reflections, I present vulnerability as source of moral responsibility. Section 5 examines the social formations that frame our perceptions and notions of vulnerability—and therefore pre-determine the scope of our responsibilities to animals. Finally, in Section 6, I turn to critique. With some justification, several scholars, such as Aaltola, have complained that relationalism is merely reconstructive, and thus relativistic [14] (p. 139). In response to this criticism, I defend a view on which social relations are open to what protagonists of the Critical Theory call “immanent critique”. Moreover, I claim that particular relationships have the potential to interrupt entrenched structures of moral consideration and animal treatment.

## 2. Moral Individualism and Some Discontents

Nicolas Delon rightly indicates that moral individualism can be understood as a formal, or meta-normative, thesis that reaches across various accounts of animal ethics, including deontological, utilitarian, contractualist, and even capability-based, accounts [18] (p. 33).[note 1] Rachels and McMahan are only two of the ethicists who defend individualism with reference to the moral and logical consistency, which is revealed in the equal consideration of individuals with comparable properties [5,6]. “The basic idea is that how an individual may be treated is to be determined, not by considering his group membership, but considering his own particular characteristics” [5] (p. 173). Both Rachels and McMahan understand human status as a purely biological fact. Presumably, they see normative dimensions divorced from this biological fact as nothing but a rhetorical ornament [6] (p. 379). On this view, the differences between different kinds of animal (e.g., companion animals, wild animals, and pests) are morally insignificant, too. Relying on the Aristotelian axiom that equals should be treated equally, both authors identify cognitive capacities as the basis of equality. Similarly, Singer’s criterion of future preferences [15] and Regan’s subject-of-a-life criterion [16] emphasize individual capacities as the basis of equal consideration. 

Some animal ethicists who emphasize vulnerability as a normative category also accept that membership of a species, or specific group, is normatively insignificant [8,9]. Martin proceeds on the assumption that an individual’s vulnerability implies that she has agency- and welfare-related interests [8] (p. 2). In her view, particular, or more precisely, heightened, vulnerability emerges when there is an increased probability of having one’s interests considered unjustly (p. 2, 11), i.e., unequally—independently of social relations. She contends that “different thresholds of wellbeing seem unjustified” [8] (p. 15), and that there is, for instance, no reason we should differentiate between a drowning puppy and a drowning wild animal. 

Moral individualism can be criticized on three interrelated grounds:

First, *capacity-oriented* individualists display Cartesian exclusivity. Although the authors actually intend to contest the implications of Cartesian dualism between body and mind—in particular, the exclusion of animals from moral considerability, as many, including Descartes, presume they lack a developed rationality or practical reason—they still treat rational capacities as having primal moral relevance. By implication, this excludes those humans and animals who do not enjoy the required level of these capacities. This is particularly visible in the frequently employed “argument from marginal cases”. There is no set of capacities or common life in which all human beings share (as some animals do) [6] (p. 362). As a result, severely cognitively disabled humans without competencies such as self-awareness allegedly have a diminished, or no, moral status, while (only) those animals which happen to have these competencies are leveled up [19] (p. 21). Giorgio Agamben unmasks this strategy as an “including exclusion” [20] (p. 37): Rational capacities (and even the “mere” ability to experience pain consciously) ensure their possessors are included in the sphere of moral consideration. They serve as criteria, while bodies and their needs and fragility are excluded or considered secondary. Therefore, some animals and some humans who do not have these capacities are regarded as beyond moral considerability. However, questions emerge: (a) Why should we adopt one criterion (e.g., having future preferences) rather than another (e.g., being able to suffer consciously)? (b) Why is this assumption of a capacity-based criteriology frequently at odds with our understanding of *special* obligations toward those who lack rationality to some degree (e.g., infants or severely impaired individuals)? (c) Are *capacities* the basis of moral responsibilities at all—or could we instead put forward a being addressed by vulnerability or, following Martha Nussbaum, a dignity of needs [21]?

Secondly, most moral individualists share the assumption that we can be purely rational, unbiased observers of the world and calculators of human and animal interests [22], although Rachels admits that this is a difficult task [5] (p. 173). Thinking and speaking from a *human* perspective, and influenced by socially conditioned *relations* to other humans and to animals (in contrast with particular relationships) is considered as anthropocentric and therefore as morally and epistemologically problematic. Similarly, it seems that the majority of individualists insinuate (albeit tacitly) that we have access to the animal per se. While in socio-cultural structures animals are present as pets, pests, precious beings, sources of nutrition, data providers, or a mere nuisance (owing to our social relations with them), these authors presume that we can “understand” animals, and their capacities and interests, divested of any social significance. In my view, this presents an epistemological and a normative problem, because, on the one hand, it obscures the fact that we understand animal capacities according to human significance (i.e., we search for and find “the human” in animals when we focus on rational capacities), and, on the other hand, our common perceptions, affective responses and practices become devalued and morally questionable *tout court*. 

Beyond individual capacities, moral individualists usually only consider particular (agent-relative) relationships, and they do so without referring to our social relations with animals [6] (p. 359) [9]. Excluding relations to animals, and emphasizing strict consistency, authors such as Rachels, McMahan, Singer, and Regan sometimes adopt what seems to be a “nagging moralistic tone” [17] (p. 469). Why should that be a problem? To further clarify this criticism, I use a telling example introduced by Martin. She indicates that when we encounter an animal at risk of drowning our obligations are the same whether it is a puppy or a wild animal [8] (p. 16). If we focus on the particular situation in which we come across a drowning wood mouse that can easily be rescued (and thus enter a particular, although very time-limited, *relationship* with the animal), it is plausible to argue that we are obliged to assist—as in the case of the puppy and as, *a fortiori*, in the case of a fellow human. However, if we consider the *social structures and relations* that sustain and predetermine particular actions of care for domestic animals—structures that are manifest in legal regulations, institutions such as official veterinary surgeon and the contents of veterinary education, and can also be detected in common intuitions regarding our responsibility for animals—we can hardly imagine that we can tailor them to distribute the same attention and care to domestic animals, liminal animals such as pigeons and rats, and truly wild animals. The lived morality and the connected social structures show a complex topography of structural recognition of individuals’ vulnerability. I argue this “recognizability” [4] (p. 4) is a result of our perspective as humans and our immersion in particular socio-cultural contexts; however; as I point out in the last section of this paper, we are not entirely determined by these structures—for, if we were, ethics, animal ethics, and every sort of critique would be futile and lack any social impact!

Thirdly, capacity-oriented individualists rely heavily on a liberalist ontology of the (non-)human subject emphasizing individuality and independence. The subject’s immersion in general relations, or particular relationships, as well as in different contexts, is either insignificant or plays only a minor role in ethical consideration. In contrast with this, Butler proposes a social ontology of the bodily self [1,4]. Each individual’s flourishing depends, first, on resources (nutrition, a decent habitat, freedom to exercise species-specific behavior, etc.). Therefore, social relations to animals are of moral significance, because there is always—as Pepper emphasizes [9]—a certain degree to which general human practices influence animal lives, habitats, and opportunities for nutrition (be it through the direct provision of resources or indirect impacts on habitats, climate, infrastructure, the food chain, etc.). Relations are those structures that determine how we frame animals as companions, pests, nutrition, etc. or as dangerous, helpful, disgusting, noble, etc.; these relations are epistemologically and practically significant. We must ask: What sorts of animal vulnerabilities are perceivable, conceivable and recognizable, and how can we tailor our practices accordingly? Secondly, in particular cases, there is a dependency on caring, or at least on non-abusive particular relationships or non-violent encounters. Palmer raises this issue in her (in my view simplistic, but instructive) distinction between a *laissez-faire attitude* to wild animals and a *caring attitude* to domestic animals [13].

## 3. Relationalism and Some Discontents

Unlike moral individualism, relationalism attempts to ground moral obligations in (the various forms of) our relatedness to animals. Following Todd May [23], we can distinguish three types of relationalism:

### 3.1. Moral (Contractualist) Relationalism 

*Moral (contractualist) relationalism* conceives of the moral community as restricted to those who can “make moral claims against one another”, i.e., humans [24] (p. 865). According to Carl Cohen, human individuals who fail to satisfy the criterion of reciprocity of moral agency (such as infants, very old people, and people with severe disabilities) are included through an argument from kind. His version of this argument can be summarized as follows:

P1: All normal human beings share morality as an essential normative feature.

P2: Infants and severely impaired individuals are humans who *actually* lack, but *potentially* share this feature. 

C: Therefore, such “marginal humans” share the same moral status as other human beings. Animals do not have a moral status as they basically lack morality. This implies that the way we are related to humans differs from the way we are related to animals [24]. 

This approach is not very attractive for animal ethics. It presupposes (at least potential) reciprocity of moral agency as basis for moral considerability, while insinuating that animals radically lack morality; such a radical lack has been contested recently in animal ethics [25]. More generally, there is no further basis for the inclusion of animals into moral consideration. Therefore, this account is not considered further in this paper.

### 3.2. Wittgensteinian Relationalism

*Wittgensteinian**relationalism*, as developed by Diamond [17,26] and Alice Crary [19,27], emphasizes the inner connection between human practices and value-laden concepts. On this view, it is shared practices that determine how we conceive and conceptualize particular *kinds* of beings (e.g., humans, companion animals, and wild animals): “We can most naturally speak of a kind of action as morally wrong when we have some firm grasp of what kind of beings are involved” [17] (p. 469). Therefore, it is not necessarily inconsistent to treat humans and animals (or different animals) differently; on this view, being human is *not* a merely biological category. May criticizes the fact that, in this account, the moral status of animals is derivative from inter-human moral relations [23] (p. 156). However, Diamond actually contends that we experience and relate to humans and animals *differently* according to social practices; animals are therefore not considered derivatively. Concepts (or “kinds”) can be traced back to these practices, and they saturate our perceptions of humans and animals. There are already normative implications (going beyond biological facts) when we perceive a human, or a companion animal, as a family member, and sewer rats as disgusting vermin or zoonotic vectors. However, both Diamond and Crary emphasize that all animals and humans are *fellow creatures* [17] or *fellow travelers in life* [19]. As I have pointed out elsewhere, they obviously prefer to employ the Aristotelian method of making this fellowship visible and addressing our sensitivity to these fellows rather than applying abstract rational principles to argue for this fellowship [28].

### 3.3. Assistance Relationalism 

*Assistance relationalism* has been put forward by Palmer [13], among others. She treats the individual’s sentience as the basis of moral considerability [13] (p. 9–25), and therefore, in my view, she is not a relationalist in the strict sense of tracing back normative obligations to relations alone (as May indeed alleges [23]). However, the nature of our differential obligations to animals is reliant on the different degrees to which each animal is dependent on humans. Palmer illustrates this by referring to the different moral intuitions we have when we consider wildebeests that drown when trying to cross waters and domestic horses lacking adequate human care. She assumes that we have a *laissez-faire intuition* in the case of wild animals (i.e., there is only a negative obligation to refrain from inflicting unnecessary harm, and assistance would be supererogatory). When it comes to the horses, however, we have a *care intuition* implying positive obligations to such domestic animals. Special obligations are derived from dependency, which is constituted by forms of contact. Humans create internal dependency (by breeding animals dependent on human care) or external dependency (by keeping animals in captivity) [13] (p. 92). Palmer identifies dependency as *realized* vulnerability; the latter concept is derived from Robert Goodin’s work, which identifies asymmetries of power as the source of vulnerability [29]. Correspondingly, individuals in power have special obligations of protection. In this respect, Palmer compares animal keeping with the decision to have a child—a decision entailing special obligations to care for offspring [13] (p. 94).

Relationalism can be criticized on four interrelated grounds:

First, relationalists run the risk of losing the individual animal (which is potentially reduced to the representative of a kind), and they could also have problems integrating the moral significance of individual capacities. In contrast, I contend that singular encounters and particular relationships (even if they are limited in time) have the potential to disrupt typical structures of relation to animals, and the roles of animals (e.g., as pests, livestock, etc.)—the individual animal subject turns out to be more than a mere example of a particular kind [30]. Drawing on Martin’s case (mentioned above) in which the drowning of a puppy and of a wild animal are compared, we could ask why we should not be obliged to assist the individual wood mouse if it is not dangerous for us to do so. Why would we distinguish in such a case between the different kinds of animal—the pets and the wild animals? Moreover, it is widely acknowledged that research findings on the rational capacities of animals have an impact on our convictions how our relations to animals should be shaped [31].

Secondly, it often remains unclear what exactly forms the identifiable kinds of animals. Neither Diamond and Crary nor Palmer give us a clue on that question. They presuppose that there is a tacit, yet shared, understanding of what a human or an animal, wild or domestic, is, and of how we relate to them; this understanding would be rooted in common practices. McMahan observes: “But what it is to be human, other than to be biologically human, is never explained” [6] (p. 371). I am nowhere near as critical as McMahan on this point, but I am willing to concede that, if we distinguish between humans and animals, and between certain kinds of animal, we have to provide clues for these distinctions—this point can made with even greater confidence if we are mindful that we have different moral obligations to different kinds of being, as suggested by Palmer [13]. The problem here deepens when we reflect on the various kinds of animal: companion animals, vermin, wild animals, liminal animals (i.e., synanthropic animals living in the vicinity of humans [32]), livestock, and so on; however, Palmer focuses mainly on the plain contrast between domestic and wild animals.

Thirdly, the question emerges why, precisely, this kind-structure has normative implications. If we construe shared practices as the only, in itself, unconditioned basis of our different obligations, we would likely conclude that they are arbitrary and/or have actually no normative implications—it is just a matter of fact that we distinguish different kinds of beings. However, as the argument of the naturalistic fallacy reminds us, we cannot extrapolate from facts to norms. Diamond and Crary do not answer the question why different shared practices are normatively significant. They do not say why, for example, such practices constitute the value-laden concept of the human—as “not something to eat”. McMahan doubts that the concept of a human being is invested with such a clear moral significance: “Yet the fact that we can coherently discuss the possibility of socially approved forms of anthropophagy (for example if they offered significant medical benefits) seems to demonstrate that Diamond’s moral commitments are not in fact embedded in our concepts” [6] (p. 376). Although I am highly critical of this assertion, I admit that he meets a point regarding the normative basis in relationalism.

Fourthly, there seems to be a presupposition in relationalism that there are *fixed* kinds of beings with fixed moral implications.[note 2] This raises a number of intricate issues: 

(a) It is doubtful whether we are entitled to assume that there are unambiguously distinct categories of animal, such as wild animals and domestic animals. If we think about liminal animals, and, for example, stray dogs or feral cats, we can see that general categories have their limits, and all the more so when we acknowledge that animals cross categories—e.g., blackbirds became liminal animals comparatively late in the nineteenth century. (b) Consider that the same species can have different social roles. Rats can be feral, liminal, or domestic animals; they are loved, used for scientific experiments, and killed if they are vermin. However, it would be naïve to assume that our social relations to rats fully determine particular relationships to animals, and that we are unaware of the differential recognition of the vulnerability of these animals. (c) As humans, we influence animal lives in far-reaching ways, even if we have no direct relationship to them, and even if this is neither intended nor actually recognized; human practices shape animal habitats and behaviors, and they might have an impact on the animals’ food resources, or change whole food chains, where invasive species that contribute to food competition, climate change, or extensive building developments are concerned [9]. Thus, can we still generally assume that a wild animal is neither interested in nor in need of human assistance, as Martin rightly asks [8]? (d) Relationalists often fail to explain sufficiently why there can be shifts in the human treatment of animals.[note 3] This implies that the resources for criticizing the current social conditions in which our relations to animals arise, and the possibilities for changing these conditions, have to be reflected more accurately.

## 4. Corporeal Vulnerability as a Source of Responsibility 

In what follows, I argue that vulnerability is a most fundamental source of moral obligations, and explain that this subverts the alleged dualism of relationalism and individualism. As noted above, the concept of vulnerability is used in remarkably different ways. Therefore, it is vital to say what understanding of this concept forms the basis of my considerations. 

Goodin [29]—and thus also Palmer [13]—conceive of vulnerability as the outcome of an asymmetry of power which should be compensated for by special obligations of those in power. Thus, they see vulnerability as a particular state that should actually be avoided, and, where it persists, places specific obligations on the part of the non-vulnerable to mitigate its effects. In contrast, in the current discourse on vulnerability within feminist ethics, poststructuralism, and phenomenology, vulnerability is understood as an inevitable part of corporeal existence [1,2,3,4,10]. Corporeality as a concept undermines the Cartesian dualism of mind and body. The focus on individual cognitive capacities, such as the “ability” to suffer consciously, is superseded by an emphasis on vulnerability as an ontological *and* normative concept. Any idea of bodily existence is accompanied and sustained by an idea of vulnerability; to understand flourishing, it is necessary to understand the possibility of being disturbed, impaired, or dying. Therefore, Martin is right when contending that human and nonhuman vulnerability are not categorically different [8] (p. 2).

Emphasizing vulnerability inevitably goes beyond a focus on the individual (although we should be mindful that we cannot but ascribe vulnerability to an individual)[note 4]. Bodies have a *social* dimension. Phenomena such as birth, parental care in many species, sexuality, cohabitation in a shared environment, and attachments show that the self-sufficient and self-determined individual is a (liberalist) fiction with, potentially, a disavowal of the shared *ontological* vulnerability of all corporeal beings. Moreover, bodies have a *public* dimension and never belong only to encapsulated individuals. Compassion and grief, in particular, show that individual flourishing and vulnerability are never only the individual’s flourishing and vulnerability. Vulnerability emerges in and through relationships, as these make us vulnerable to grief and *com-*passion (to suffer-with), as well as to exploitation, violence, instrumentalization, etc. In short, therefore, the inevitable proximity to others is ambivalent: “[T]he skin and the flesh expose us to the gaze of others, but also to touch, and to violence” [1] (p. 26). 

This implies that vulnerability has normative significance. Gilson rightly asserts that “an idea of vulnerability underlies our notions of harm and well-being, interests and rights, equality and inequality, and duties and obligations” [3] (p. 15). It is no coincidence that several authors involved in current debates on vulnerability [1,4,10,33,34] draw on the writings of Emmanuel Levinas—whose work has been fruitful in animal ethics, too, even though he was rarely concerned with morality in the face of animals [14,35]. The fundamental idea in Levinas’ oeuvre is—as Aaltola expresses so well—that vulnerability “gives birth to ethics” [14] (p. 131). Vulnerability is not only passivity, weakness, and fragility but also something with the power to address us [36] (p. 49). Levinas calls this the *saying* (of the face), which is *not* connected with the capacity to talk;[note 5] the vulnerability of the body (signified by the concept of the face) is what *addresses me*, and that makes it impossible for me not to respond to (the claim of) the other. Jacques Derrida has analyzed such a being addressed by animals using the example of the encounter with a cat; he complains that “Levinas never … evoke[s] the gaze of the *animot*[note 6] as the gaze of that naked and vulnerable face to which he has dedicated so many beautiful and gripping analyses” [37] (p. 107). Ignorance is—similar to assistance or violence—a response, and it therefore has moral significance. If I merely pass by a beggar asking for money, a hedgehog (a liminal animal) trapped in a fence, an injured cat (a companion animal) on the sidewalk, or a wood mouse (a wild animal) that is about to drown, then that is a response to the—in these cases not only ontological, but also situational and heightened [2] (p. 7f.)—vulnerability of the other. 

Levinas asserts: “The best way of encountering the Other is not even to notice the color of his eyes” [38] (p. 85). In saying this, he is signaling that particular capacities or other characteristics (e.g., species membership)—*prima facie*—do not play a role when it comes to ethics. An infant, individuals in a coma, but also—with and against Levinas—animals (even “lower” animals) are therefore addressing us in their appearance.

However, our responsibility to vulnerable beings, as I argue in the next section, is inevitably differential, and this turns moral responsibility into a complex and intricate issue. The recognition of vulnerability is dependent on relations (i.e., social structures and frames that determine the visibility of vulnerability as well as our affective responses and dispositions to act), and it happens within particular relationships and occasional encounters. We do not experience an abstract vulnerability per se. Consequently, we do not take every sewer rat to the veterinarian (although there is an acknowledged obligation to give—a however differential—veterinary care to rats kept as pets or in laboratories). Although Martin [8] is right to indicate that we should be as responsive to a wild animal as we are to a companion animal—e.g., in the particular situation where the animal is about to drown (because a particular encounter never merely reproduces social relations [30])—the overall normative infrastructure of our lifeworld inevitably creates differences. 

## 5. Recognizing Animals’ Vulnerability

It seems clear that my responsibility differs depending on whether I am facing my own children or other children, my dog or other dogs, or my pet rat or sewer rats. Delon rightly asserts that there is a special wrongness in neglecting or harming my children or my dog that is due to our special relationships [18] (p. 40), and it would be a special form of wrongness to prefer my dog to my child, even if she is an infant of one week lacking self-awareness and the dog is already self-aware. Against moral individualism, I contend that our moral life is properly mirrored, not in the unbiased application of criteria for equal consideration of biological equals, but in our differential—although not arbitrary—recognition and responsiveness to various forms of vulnerability. 

Wittgensteinian relationalists such as Diamond and Crary would explain this by referring to the (normative) implications of the concept “my child” in contrast to “my” or “any dog”; they would say that these implications are constituted by shared practices. In a different vein, we can explain such differences as instances of “reasonable partiality” [18] (p. 40), which can be traced back to broader *social relations.* The ways in which we, as humans, relate to humans, and to different kinds of animal, are determined prior to the emergence of *particular relationships and encounters*. These relations constitute a social structure, which forms the basis of the reasonableness of the partialities. Butler uses the (epistemological and value-laden) notion of *frames* and the complementary notions of recognizability and grievability to conceptualize this structural, *historical* (i.e., contingent) *a priori* [4] (p. 1, 4). It is not primarily or only the “simply” man-made dependency of children (arising from the decision to reproduce) or of animals (arising from the decision to breed and keep them) that constitutes special responsibilities (as Palmer alleges). To be sure, dependency is crucial for special responsibilities; however, causally induced dependency alone does not fully determine particular responsibility. This becomes clear if we take again Martin’s example of drowning individuals: if I see my own puppy and a (perhaps even severely cognitively disabled) child I have never met before drowning, I am expected to save the child first and foremost. Saving the puppy first, with the risk that the child will die, would be regarded as deviant or inhumane behavior. Social relations determine how particular relationships are configured and what kinds of responsibility emerge from them. As Butler puts it, “there is no life and no death without a relation to some frame” [4] (p. 7). Children and dogs are differently framed as vulnerable beings, and moral diversities become even more visible when we think about the differences between animals regarded as pets (family members), livestock animals, feral animals, and animals identified (framed) as pests. The last of these might even be to some extent inconceivable as vulnerable beings that evoke responsibility and merit respect or even assistance. In Butler’s terminology, we can agree that sewer rats have very limited grievability. “The apprehension of grievability precedes and makes possible the apprehension of precarious life. Grievability precedes and makes possible the apprehension of the living being as living, exposed to non-life from the start” [4] (p. 15). However, moral behavior is responsive, not to social relations or frames, but, precisely, to an *individual’s* vulnerability. As shown in the next section, the social relations framing vulnerability are powerful, yet not fully determinant, and thus susceptible to change, not the least through the responsiveness to individuals.

This should lead us to query McMahan’s assertion that relations are less morally significant than intrinsic rational capacities [6] (p. 354), since vulnerability cannot be mistaken for a rational capacity, nor can it be understood independently of social relations or frames. When we understand responsibilities solely as the outcome of shared practices (and concepts derived from these practices), we readily assume they are arbitrary; we then face the very same problems as Wittgensteinian relationalists. This is explosive when we consider moral individualists’ attempt to object to the differential consideration and treatment of humans and animals on grounds of cognitive capacities; their demand for moral and logical consistency would be powerful (and in some instances, it amounts to a strong argument). However, in my view, some of these differences—although by no means all—are neither arbitrary nor, strictly speaking, irrational, because the social relations of recognizability and grievability pre-determine the scope of responsibility and constitute a topology of moral considerability, which, in turn, introduces schisms [4] (p. 50) in our responsiveness to animals. Against this background, vulnerability becomes, as Gilson puts it, a “complex socially mediated phenomenon” [3] (p. 7). Different socially determined relations to animals (preceding particular relationships to individual animals) constitute the reasonableness of “reasonable partiality” [18] (p. 34). There is a difference in the perceivability of vulnerability, and connectedly differential affective responses and dispositions to act precede any avowal or disavowal of our moral obligations to animals.

Crucially, human agents—as with animals—are to be understood as corporeal beings. Corporeality implies being related to others and immersed in a particular socio-cultural context. Practical reason is not a free-floating capacity, which is accidentally connected to a body (as it is depicted as being in Cartesian dualism). We are embodied beings who are—as knowers, thinkers, and actors—deeply influenced by social structures of significance and of recognition, and a social economy of emotions. The French philosopher Maurice Merleau-Ponty has analyzed the implications of a radical thinking-out of corporeality for our understanding of rationality and intentionality. As bodily beings, we have a particular perspective on the world [39] (p. 73). This is not only the case as regards our sensual perceptions, but also when we consider a problem from “different angles” [40] (p. 121). Moreover, reason is sustained by a sub-intentional arch of bodily skills and dispositions—we do not stumble from one encapsulated situation to another; our dispositions to perceive and act allow us to be oriented in most of the situations we are in [40]. As social beings we have socio-culturally imbued dispositions to perceive animals, to experience affective responses to them, and to treat them in certain ways. I am disposed to a particular, perspectival form of openness to vulnerability. Animal encounters are therefore, first and foremost, *framed* encounters [30]. I perceive a human, a companion animal, or a livestock animal according to a social normality and connected normative structures. Practical reason is therefore never entirely unbiased, and the individual actor never takes the “point of view of the universe”. Animals are never visible as animals per se—and humans are not just members of a particular species; all living beings turn out to be framed. 

To explain the concept of a frame, I paraphrase Charles Rosenberg’s famous reflections on disease in a way that makes them fruitful in our context [41] (p. xiii): *An animal is at once a biological entity; a generation-specific repertoire of verbal constructs reflecting our relations to the animal; an occasion of, and potential legitimation for, public policy;, an aspect of social role and individual identity; a sanction for cultural values; and a structuring element in human–animal interaction.* The same is true of humans. Therefore, our attempts to tackle the animal question, and to reflect ethically on our treatment of animals, always happen against the backdrop of the question who we are. We are not distinguished from animals (something that could be grasped only from the disembodied, dissocial point of view of an impartial observer) but we distinguish *ourselves* from *them*; a human self-understanding can never be detached from our conceptions of (different) animals. Therefore, I consider Rachels’ assertion as somewhat beside the point: “It has always been difficult for humans to think objectively about the nature of non-human animals” [5] (p. 129). It is impossible; however, this does not imply that we cannot be sensitive to the fact—as noted above, drawing on Martin [8]—that human and animal vulnerability are not categorically different. Nevertheless, the frames that constitute humans as humans and animals with different kinds of social significance pre-determine the scope and limits of our perception of, responses to, and compassion for, other living beings and their vulnerability.

## 6. Trapped in Frames? Sources of Critique

Authors such as Rachels, McMahan and Singer tend to presuppose that we are able to treat humans and animals solely in accordance with their mental capacities. However, first, it is questionable whether we can safely assume that individual capacities build the basis—and perhaps the only basis —of moral considerability. Secondly, I contend that we are bodily beings who are deeply influenced by social relations, which are neither easily nor entirely at our disposal. Thirdly, it seems seriously doubtful that a social and moral life is possible without a differential structure of recognition. Recognition for everyone and everything ceases to be recognition at all. To illustrate this thought: The idea that we can be attentive to everyone and everything is oxymoronic, since attention builds an enhanced figure against a horizon. Similarly, recognition of vulnerability without a topology would appear to be impossible, since in that alleged kind of recognition we would either lack any criterion to distinguish between the vulnerability of a baby and that of smallpox or we would get trapped in the same aporias—similar to how some individualists have difficulties acknowledging the complexity of morally significant social relations.

However, we have to ask how an approach to animal ethics relying on the social structures surrounding the recognition of vulnerability can be more than merely reconstructive. Thus, what sources for a critique of current conditions are available to such a theory? Pelluchon rightly interprets Derrida’s neologism *animot* as expression of the fact that we and our relations to animals tend to be caught up in concepts and—as I would like to add—connected frames of perception and affective response [7] (p. 6). As corporeal beings, we perceive, think, and act from a particular perspective and against the backdrop of social relations that are ingrained in our habitual dispositions. We tend to perceive and respond to animals in accordance with these social normalities. We are thus also vulnerable to being trapped in frames: “[T]he skin and the flesh expose us to the gaze of others, but also to touch, and to violence, *bodies put us at risk of becoming the agency and instrument of all these as well*” [1] (p. 26, emphasis added).

In the vein of Butler and Derrida, I want to defend a view on which we are entirely “bewitched” neither by language nor by social frames of recognition. Conceptions of humans and animals, as well as frames, pre-determine how we perceive and conceive of living beings, yet they are contingent, complex, and to a certain degree open to critique from within social relations. 

The issue of contingency (i.e., changeability) is trivial from a historical perspective; by the same token, it reveals the ambivalence of this contingency. Since the 1970s, it has been possible to sustain animal ethics as an academic discipline. Presumably, this would not have been possible some decades previously. Moreover, the treatment—in actual fact, the recognizability and treatability—of animals has changed significantly. While some animal bodies have become reified in an unprecedented way (in factory farming), companion animals have become veritable family members introducing previously unthinkable levels of moral respect, as can be witnessed in, for example, the highly sophisticated veterinary treatments for pets now being offered (renal transplantations in cats, blood donations in cats and dogs, etc.).

Within this complexity of social relations, different relations reveal possibilities of different treatments for animals. Instead of interpreting the differential conceptions and treatments of animals (as family members, resources, vermin, etc.) as mere logical and moral—and inexcusable—inconsistency, we can explore the idea that this could be a basis for what Critical Theory has termed *immanent critique* [42] as basis for moral change. Drawing on Marx, the basic idea is that this form of critique gives expression to already existing, yet implicit, normative conflicts within social practices [42] (p. 3). Where the treatment of animals is concerned, these conflicts arise as a result of the diversity and complexity of frames. It is not at all obvious how, exactly, we are to treat livestock animals, feral animals, or companion animals. Moreover, the various practices within the many forms of human–animal relation reveal alternatives for each particular field. Therefore, practices such as meat consumption, factory farming, and animal experimentation do not go uncontested in our lifeworld. It seems clear that these practices do offer alternatives. However, this does not come about through the application of abstract principles from the point of view of an impartial observer.

Occasions to call structural relations into question are particular experiences. As noted, frames are powerful and saturate our perceptions of humans and animals—our affective responses as well as our dispositions to treat them in the ways we do. This is due to our existence as finite, bodily, vulnerable beings that are embedded in social relations and dependent on their bodily habits. Social relations are therefore inert and display a certain gravity. Thus, I do not see how we can construct a *total* critique; we are incapable of taking the point of view of the universe and looking at matters from nowhere. However, animals are never “entirely” framed. A livestock animal can hardly be conceived *merely* as a resource; any reification of animals must ultimately fail in view of the subjectivity and vulnerability (of the individual animal), which we can experience in particular relationships and encounters. Frames can become open to question. Butler indicates that “to call the frame into question is to show that the frame never quite contained the scene it was meant to limn, that something was already outside, which made the very sense of the inside possible, recognizable” [4] (p. 9). Butler uses the concept of “apprehension” to make this visible:
“What we are able to apprehend is surely facilitated by norms of recognition, but it would be a mistake to say that we are utterly limited by existing norms of recognition when we apprehend a life. We can apprehend, for instance, that something is not recognized by recognition. Indeed, that apprehension can become the basis for a critique of norms of recognition.”[4] (p. 5)

Finally, as Delon rightly points out, individual capacities can affect relationships [18] (p. 42). Take the example of cognitive biology in the twentieth century. Until some point after World War II, it was commonly believed that humans can be distinguished from animals with reference to properties such as rationality or reason, language, self-awareness, culture, tool-use, etc. However, in the past few decades, this kind of distinction between us and them, cast in terms of capacities, has increasingly been seen as an allegation rather than a fact. Our knowledge of the capacities has heightened our sensitivity to animal subjectivity—and vulnerability—as can be seen from the establishment of animal ethics as academic discipline, public discourse, and changes in animal protection laws. However, this knowledge alone does not entirely change our relations to animals, nor does it alter the social significance of animals *tout court*. As noted, the recognizability of animals’ vulnerability is inert and never fully at our disposal—yet it is open to an ongoing critique. 

## 7. Conclusions

In this paper, I argue that the concept of vulnerability can be conceived in such a way that it undermines the alleged opposition between moral individualism and relationalism. I proceed from an ontology of the bodily subject as dependent on the sustenance of flourishing; vulnerability is an inevitable, pervasive condition which humans and animals share. Crucially, it is not to be understood merely as a characteristic of an individual who can be detached from the *social relations* in which she is immersed and the *particular relationships* in which she is embedded. While the former constitute and represent frames, which are a condition of the intelligibility of vulnerabilities, and of affective and moral responses to vulnerabilities, the latter are opportunities in which we are directly addressed by vulnerability and which potentially thwart the structure of recognition. Although the recognizability of vulnerability remains the horizon of responsibility, particular relationships and encounters as well as immanent critique are possible elements of critical inquiries into our lived morality.

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
