# Peer review of "How to Recognize Animals’ Vulnerability: Questioning the Orthodoxies of Moral Individualism and Relationalism in Animal Ethics"

_animals, 2020, doi:10.3390/ani10020235_

Round 1

Reviewer 1 Report

This is an excellent paper that reflects the latest state of research and offers an original and important contribution.

The critique of moral individualism and relationalism is just about the right length to show what is at stack in the debate before proceeding to the authors positive proposal. The particular strength of the paper is the nuanced discussion of vulnerability as source of normativity in which the author also addresses different pitfalls in resorting to this concept. The final two parts, linking the concept of vulnerability with matters of recognition and social critique, demonstrate the complexity of human animal relations. The approach defended in this paper promises to be better capable at addressing that complexity than traditional accounts in animal ethics. 

It is a pleasure to read the paper. The argument is clearly structured, the writing style is easily accessible and precise at the same time. All concepts are appropriately introduced and clearly defined. Thus, I recommend publication.

Reviewer 2 Report

This paper is a good review and critique of both moral individualist and relational approaches to animal ethics. It offers a new approach to relationalism based on creatures' vulnerability. This approach draws on the work of poststructuralism and phenomenology and the writings of thinkers such as Butler, Mackenzie, and Gilson, as well as those of  Bourdieu and Levinas. This work has not been previously applied to animal ethics, as it is here. The value of this paper is that it establishes the vulnerability position within animal ethics.

This position is not without its own problems and questions. As the author himself points out, "it is doubtable whether ethicists can rely on the extraordinary experience of an encounter that exceeds ordinary structures of treating animals." In general, it remains unclear what forms of normative guidance operates within and without the fields and frames that shape (or critique) the recognition of vulnerability.

On a different note, it might be asked whether the dismissal of moral individualism on the grounds of its exclusion of human infants is well founded. Kant, the preeminent rationalist and moral individualist, founded our duties both to young children and animals on the reflexive impacts on autonomous rational moral agents themselves. This approach not only merits continuing attention in its own right, it may provide some of the (lacking) normative apparatus for the vulnerability position itself.

These points merit development and response in independent responses to the vulnerability position staked out here. They are not meant as criticisms of the paper itself nor as revisions the author has to make, although in some form they may warrant  inclusion as cautions or qualifications offered by the author.

Reviewer 3 Report

The author seeks to defend a middle ground between moral individualism and relationalism, arguing that vulnerability is the fundamental source of obligations, but that it does not predetermine our behavior, attitudes and practices so that it would preclude their critique. I happen to be sympathetic to attempts to bypass the false choice between either capacities or relationships as exclusive grounds for moral status. I don't think the author can pull it off, however, for a number of reasons.

First, they overlook pertinent sections of the literature. I am not mentioning this lightly, as calls for citing the relevant literature are often overblown, but in this specific case this affects directly the originality and significance of the author's proposal. Thus, it does not bode well when the author writes (p. 2, l. 56-58):

we find only a few scattered reflections on animals within the established discourse on vulnerability and only a small number of scholars who have yet taken up this concept [of vulnerability] to use it within the context of animal ethics

Most of care ethics comes to mind, but I also recommend having a look at recent work by Elisa Aaltola, Nicolas Delon, Angela Martin, Corine Pelluchon, or Angie Pepper. All have published on the topic. Clare Palmer, discussed at some length in this paper, has also said more about animal vulnerability than is covered here.

On moral individualism, the author mainly seems to rely on the characterization they took from its critics. Its most prominent and explicit defender, James Rachels is not discussed, merely cited. McMahan, another prominent moral individualist and a critic of Wittgensteinians, is not even cited. I don't think these discussions can be fruitfully approached without at least briefly mentioning these existing debates.

Secondly, while many of the author's point are well taken, the core steps of their argument fails to convince, mostly because it does not rest on argumentation. So many assumptions are either unstated or taken for granted, and more often than not the author takes the word of other authors, such as Bourdieu or Butler, as authoritative. But citing them, has never had, by itself, any evidentiary value in an argument. The author wants to defend a view that they don't seriously want to be arguing for.

So, I would suggest that the author clearly identify the argumentative steps required to support the claim that vulnerability is the source of moral obligations to other creatures. The author reproaches relationalists for not explaining the normative basis of their views, but the author's argument falls prey to a similar criticism.

Round 2

Reviewer 3 Report

The author has substantially improved the paper in response to feedback. It would be a nice addition to the special issue and of interest to animal ethicists and readers of Animals.

Author Response

Hi,

I wanted to thank you once more for your initial comments on my text. I really could improve the argumentation and I could, for myself, better understand my own position!

And - of course - thank you for the positive feedback on the revised version!

Best wishes

Martin